# Arsenic Contents, Speciation and Toxicity in Germinated Rice Alleviated by Selenium

**DOI:** 10.3390/foods12142712

**Published:** 2023-07-15

**Authors:** Xin Zheng, Jing Hong, Jingyi Zhang, Yulong Gao, Peng Li, Jian Yuan, Guanglei Li, Changrui Xing

**Affiliations:** Key Laboratory of Grains and Oils Quality Control and Processing, College of Food Science and Engineering, Collaborative Innovation Center for Modern Grain Circulation and Safety, Nanjing University of Finance and Economics, Nanjing 210023, Chinazhangjingyi221@163.com (J.Z.);

**Keywords:** HPLC-ICP-MS, As(III), As(V), Se, rice, germinated

## Abstract

Rice can accumulate more organic and inorganic arsenic (iAs) than other crop plants. In this study, the localization of As in rice grains was investigated using High Performance Liquid Chromatography-Inductively Coupled Plasma Mass Spectrometry (HPLC-ICP-MS) based on 26 rice varieties collected from two provinces. In all the samples, the total As contents in polished rice were 0.03–0.37 mg/kg, with average values of 0.28 and 0.21 mg/kg for two sample sets. The results of the determination of arsenic speciation in different components of rice grain showed that in the polished and brown rice the mean value of arsenite (As(III)) was nearly twice than that of arsenate (As(V)). The regional difference was observed in both total As contents and As speciation. The reason may be that As(III) is more mobile than As(V) in a dissociated form and because of soil properties, rice varieties, and the growing environment. The proportion of iAs and the total As in rice bran was higher than that in polished rice, and this is because As tends accumulate between the husk and the endosperm. In our study, selenium could alleviate the risk of arsenic toxicity at the primary stage of rice growth. Co-exposure to As and Se in germinated rice indicated that the reduction in As accumulation in polished rice reached 73.8%, 76.8%, and 78.3% for total As, As(III), and As(V) when compared with rice treated with As alone. The addition of Se (0.3 mg/kg) along with As significantly reduced the As amount in different parts of germinated rice. Our results indicated that Se biofortification could alleviate the As accumulation and toxicity in rice crops.

## 1. Introduction

Inorganic arsenic (iAs) is one of the top ten chemicals endangering global public health and has been listed as a Group I human carcinogen by the International Agency for Research on Cancer (IARC) [1]. Studies have shown that low levels of inorganic arsenic exposure in drinking water and rice are associated with the risk of myocardial infarction and kidney diseases [2,3]. Because of natural processes and various anthropogenic activities, such as the weathering of rocks and the use of As-containing pesticides and herbicides in crop production, arsenic is widely present in rice through arsenic transformation in flooded paddy soil [4].

Because of the interactions between the rice rhizosphere and the arsenic in paddy soil and the methylation process, the content and speciation of arsenic in rice are complicated and have been the focus of attention as rice consumption is the major route of As exposure to humans [5]. A study showed that the total As contents were 25–327 μg/kg based on the analysis of 108 rice samples collected from 13 major rice-producing regions in China, where iAs was the dominant species [6]. An investigation from Italy showed that the mean total As content of 168 white rice samples was 155 ± 65 μg/kg, with the mean iAs 102 ± 26 μg/kg [7]. A total of 1180 polished (white) rice samples were analyzed from 29 distinct sampling zones of six continents with the mean inorganic arsenic 66 μg/kg and dimethylarsinic acid (DMA) 21 μg/kg [8]. The As contents showed significant differences among producing areas, and the inorganic As was largely below the E.U. limit of 150 μg/kg for polished or white rice because of the As transformation [9]. However, most investigative attention has been focused on the concentration of inorganic arsenic in the polished rice. The distribution of the inorganic arsenic and organic arsenic (also as organically bound As) within the cereal grain will aid understanding of their location patterns and the impact of processes such as milling.

Because silicon (Si) is especially essential for *Oryza sativa*, more than 10% of Si in the dry weight of the leaves and husks was accumulated to ensure the high and stable production of the paddy. Genes responsible for Si transport (*OsLSi1* and *OsLSi2*) and PO_4_^3−^ were found to be involved in the process of As(III), As(V), methylarsonic acid (MMA), and DMA uptake from soil to grain. Si accumulation by Si fertilizer application can decrease As accumulation for rice plants in arsenic-contaminated soils.

In order to control the As accumulation in rice plants, much research has found that some sulfur-containing compounds, such as glutathione and phytochelatin, are responsible for As complexation, and their existence affects As uptake, translocation, and accumulation. Through the complexation of As(III) by glutathione and phytochelatins, this mechanism could reduce As translocation to rice grain. Kumarathilaka P et al. showed that As(III) translocation efficiency could be limited by its complexation with thiols [10]. In rice roots, the formed As(III)–thiol complexation influences the accumulation of total As in rice grains. At the same time, different MMA(V)–thiol complexes were also found in the rice shoots and roots. A. K. Srivastava et al. reported that the supplementation of thiourea to As(V) reduced the As concentration by 56% in the aerial parts through the significant downregulation of the Lsi2 transporter [11] and the upregulation of sulfate transporters. This supplementation increased the activities of sulfur assimilation-related enzymes and then partially ameliorated the effects induced by As stress.

Recently, there has been research with respect to Se treatments for the accumulation of As in grain. Nanoscale secondary ion mass spectrometry (NanoSIMS) analysis showed that Se and S have similar distribution patterns, indicating these two elements are transported and deposited in the grain based on the same pathway [12]. By antagonizing the toxic effects of As, Se treatments have shown a mitigation effect on plant height, leaf dry weight, and grain yield [13]. Some exposure assessments revealed that Se biofortification could alleviate the As accumulation in rice [14] and significantly mitigate the harmful effect of As on the reduced yield of rice grain [15]. However, a study found that Se fertilizer treatment increased Se concentrations greatly but had no obvious effects on the concentrations of Cd and As in brown and milled rice [16]. Although some contradictory results were obtained, sulfur and selenium biofortification to decrease heavy metals in plants is a worthwhile strategy in rice production. Recently, Fang-jie Zhao et al. identified that the gain-of-function *arsenite tolerant 1* (*astol1*) mutant of rice was related to enhanced sulfur and selenium assimilation [17]. Based on the previous findings, an evaluation of selenium assimilation associated with the mitigation of As in germinated rice was conducted in this paper.

In this study, total As content and the localization of As speciation in rice were investigated using HPLC-ICP-MS to analyze 38 samples from two rice-producing provinces. The distribution patterns of As from the two areas were compared. To evaluate the effect of the Se treatments on the accumulation of As in rice, the effect of the interaction between As and Se in germinated rice was evaluated to better understand the mitigation effect of Se on As contents and speciation.

## 2. Materials and Methods

### 2.1. Chemical and Standards

Nitric acid (HNO_3_, 65%, guaranteed grade) was purchased from Kermel Chemical Reagent Co., Ltd. (Tianjin, China). Ultrapure water with a resistivity of 18 MΩ cm was obtained from a Millipore-Q system (Bedford, MA, USA). Arsenic acid (As(V)), arsenious acid (As(III)), methylarsonic acid (MMA), dimethylarsinic acid (DMA), and selenite were provided by Aladdin Reagent Co., Ltd. (Shanghai, China). Certified reference material for the chemical composition of rice flour was provided by NCS Testing Technology Co., Ltd. (Beijing, China).

### 2.2. Sample Collection and Preparation

In this study, a total of 38 rice samples were collected from local markets from 2 rice-producing provinces. The raw rice material used for germination experiments was harvested in Jintan District, Changzhou City, Jiangsu Province.

Rice hull, rice bran, brown rice, and polished rice samples were prepared according to the following steps. First, dust and impurities in the paddy were eliminated by a circular sieve with a diameter of 2.0 mm. Subsequently, the JGMJ8098 rice huller (Shanghai Jiading Cereals and Oils Instrument Co., Ltd., Shanghai, China) was employed for the husking process of the rough rice to obtain brown rice and rice hull. The distance between two rubber rollers was adjusted depending on the size and morphology of the rice grains. The JNMJ6 rice milling machine (Zhejiang Taizhou Grain Instrument Factory, Zhejiang, China) was used to produce polished rice from brown rice. The simultaneously yielded rice bran was also collected. Finally, the JXFM110 hammer-type cyclone mill (Shanghai Jiading Cereals and Oils Instrument Co., Ltd., Shanghai, China) was utilized to crush the rice hull small enough to ensure that more than 95% of the sample powder could pass through a 40-mesh sieve.

### 2.3. Total Arsenic Analysis

The As in rice hull, rice bran, brown rice, and polished rice was extracted by the microwave-assisted extraction method [18], and the parameters of the method are summarized in Table 1. Specifically, 0.300 ± 0.001 g of the sample was weighed and digested by the addition of 3 mL of 65% HNO_3_ and 2 mL of 30% H_2_O_2_ in a microwave digestion tube. Then the samples were treated according to the conditions listed in Table 1 by microwave digestion system. The obtained clear digestion solutions were evaporated by an electric heater at 160 °C. The residue was transferred to a volumetric flask and fixed with 2% HNO_3_ solution to 10 mL. A reagent blank was used as a control. The content of arsenic in samples was tested by ICP-MS.

### 2.4. As Speciation Analysis

For As speciation, dilute HNO_3_ was used to extract inorganic arsenic and organic arsenic from the rice hull, rice bran, brown rice, and polished rice samples. The process followed the GB 5009.11—2014 method according to the previous method [6]. The heat-assisted acid extraction method was adopted. Powdered rice samples of 1.000 ± 0.001 g were weighed and placed into a 50 mL polypropylene centrifuge tube. A total of 20 mL of 0.15 mol/L HNO_3_ solution was added and the mixture was allowed to stand overnight. Then, the samples were extracted at 90 °C for 2.5 h and shaken every 0.5 h for 1 min. The solution was cooled to room temperature and centrifuged at 8000 rpm for 10 min. The supernatant was filtered with 0.22 μm filters. The samples were stored at −80 °C before speciation analysis. Arsenic species were determined using HPLC-ICP-MS. The parameters of HPLC-ICP-MS for arsenic species determination are shown in Table 2. At the same time, the reagent blank was prepared according to the same method.

### 2.5. Effect of Selenium Application on Arsenic Uptake in Germinated Rice

Selenium at lower concentrations is reported to ameliorate the As stress on rice seed germination [19]. The application of selenium to alleviate the risk of arsenic toxicity was evaluated by the germination process. Germinated rice was prepared by seed germination equipment at a controlled temperature (~25 °C). The seeds were germinated in water for 3 d. Se additions in water of 0, 0.3, 0.6, 1.2, and 2.4 mg/kg were used at the germination process with the same levels of arsenite:(1)Control: 3 mg/L arsenite without selenium addition;(2)Se0.3: 3 mg/L arsenite with addition of selenium at 0.3 mg/kg;(3)Se0.6: 3 mg/L arsenite with addition of selenium at 0.6 mg/kg;(4)Se1.2: 3 mg/L arsenite with addition of selenium at 1.2 mg/kg;(5)Se2.4: 3 mg/L arsenite with addition of selenium at 2.4 mg/kg.

Then the germinated samples were dried and stored at −4 °C. Finally, the paddy was treated into the rice hull, rice bran, brown rice, and polished rice samples. The total As and As speciation were detected as described before.

## 3. Results and Discussion

### 3.1. Total Arsenic Content in Different Parts of Rice Grain

Total As contents in rice samples were arranged based on the sample resources. A total of 15 rice varieties (numbered 1–15) were arranged first, including 19 rice samples from one province, where variety 1 had two rice samples and variety 9 had four rice samples. The remaining 11 rice varieties (numbered 16–26) were arranged, including 19 rice samples from another province, where varieties 16 and 18 both had two rice samples and varieties 19, 20, and 22 had three rice samples.

Certified reference material for the chemical composition of rice flour was used to validate the established analytical method. The average value for two certified reference materials were 0.19 and 0.13 mg/kg, with 3.0% and 4.3% relative standard deviations (RSD, %).

As shown in Table 3, the total As contents in the polished rice samples were 0.03–0.37 mg/kg, with average values of 0.28 and 0.21 mg/kg for samples from two provinces. Our data were consistent with Jia-Yi Chen et al. who reported that the total As contents in 108 rice samples was 0.025–0.327 mg/kg (0.120 mg/kg). In our results, all polished rice samples for rice varieties 1–15 showed a high concentration of total As beyond 0.2 mg/kg [20]. The variable total As contents in rice and brown rice were observed [21]. Hongping Chen et al. found total As contents at 0.011–0.186 mg/kg and Nookabkaew et al. determined brown rice in Thailand to have a higher total As content (0.118–0.346 mg/kg) [22,23]. WHO has set a permissible limit for iAs of 0.4 mg/kg for brown rice, and for all the brown rice samples of rice varieties 16–26 the total As content was lower than this maximum limit. The concentration of total As in the powdered samples successively increased from the polished rice and brown rice to the bran and husk.

### 3.2. Determination of Arsenic Species in Different Parts of Rice Grain

Four arsenic species were detected by HPLC-ICP-MS. These four species existed in rice samples with high frequency. Arsenic intake through rice in China is higher than from drinking water, with a 37.6% contribution to the maximum tolerable daily intake (MTDI) of As [24]. Each compound contained six concentration levels (1, 5, 10, 20, 50, and 100 ng/mL). Peak areas were used for quantification as shown in Figure 1, and the linear was very well defined for each compound. Correlation coefficients were higher than 0.992 for all species. The obtained parameters were similar to those reported by M. Bissen et al. [25]. The analytical performance for determination of arsenic species was studied before by our group [18].

The results of the determination of the arsenic species As(III), As(V), MMA, and DMA in different components of the paddy are shown in Table 4. No MMA was found in any of the samples. This was consistent with the results in which MMA was below 10 μg/kg in all the analyzed rice, rice crackers, rice-based infant cereal, and corn-based infant cereal [26]. Although the amount of MMA is usually low or negligible [27], some MMA could exist in the husk, where inorganic and organic species have an equal proportion(%), as assessed by X-ray absorption near-edge spectroscopy [28]. The low MMA revealed by our data was consistent with the reports of Jia-Yi Chen et al. [6] and Meharg et al. [29].

In the polished and brown rice, the sum of the mean values of As(III) and As(V) ranged from 0.045 and 0.19 mg/kg, in which the concentration of As(III) was found to be the nearly twice that of As(V) in the samples from both areas. The results of Jia-Yi Chen et al. also indicated that the As(III) was found to be the major component [6]. This phenomenon was related with arsenic species in soil. Much research has shown that As(III) is more mobile and toxic than As(V), mainly in a dissociated form [30]. Arsenic mobility in soils was associated with the species and the type of soil, and As(V) is considered of low mobility and more strongly retained in the soil [31].

In Figure 2**,** As speciation in rice hull, rice bran, brown rice, and polished rice from 26 varieties (38 samples) is shown. In Figure 2A, the concentrations of As(III) in rice hull and rice bran were comparable except in four varieties (5, 6, 7, and 10). The average concentrations of As(III) in the polished rice from two provinces were 0.12 and 0.099 mg/kg, which were lower than that in bran about 4- to 5-fold. As shown in Figure 2B, the concentration of As(V) was higher in rice bran than that in rice hull, brown rice, and polished rice. In rice hull, the concentration of As(III) was obviously higher than As(V) and DMA. However, in rice bran, the content of As(V) was lower than the content of As(III) and DMA. The average concentrations of As(V) in polished rice were 0.070 and 0.045 mg/kg, which was the lowest among the three detected As speciations. The results were consistent with an investigation of the amount of As(III) in a global suite of 53 rice brans, which was higher than that of As(V) [32]. A balanced presence of As(III): As(V) speciation shown by the 3C spectrum of arsenic (As) µ-X-ray absorption near-edge spectroscopy spectra also indicated this phenomenon [28]. The sum of As(III) and As(V) in rice bran from Thailand was previously found to be around 0.6 mg/kg by Ruangwises et al. [33]. However, the total value was very low in some samples (0.017 mg/kg from Guatemala and 0.021 mg/kg from Madagascar), indicating the As in rice bran varied widely according to the global source [33]. In general, the proportion of iAs and the total As in rice bran was higher than that of polished rice, and this is because there is evidence that As is mainly localized between the husk and the endosperm, such as the aleurone and outer parts of the endosperm [28].

### 3.3. Total Arsenic and Speciation in Co-Exposure As and Se in Germinated Rice

In order to investigate whether selenium could alleviate the risk of arsenic toxicity at the primary stage of rice growth, we focused on total arsenic and its speciation changes in co-exposure to As and Se in germinated rice. In general, the total arsenic, As(III), and As(V) in grains were decreased in the presence of Se for all Se added groups (0, 0.3, 0.6, 1.2, and 2.4 mg/kg) in rice hull, rice bran, brown rice, and polished rice, as shown in Figure 3. In rice hull, the migration concentration was decreased from 15.1 to 4.7 mg/kg for total As content, from 7.7 to 3.2 mg/kg for As(III), and from 7.9 to 2.4 mg/kg for As(V). In rice bran, the detected total As content ranged from 16.4 to 5.4 mg/kg and the corresponding As(III) and As(V) ranged from 11.1 to 3.4 mg/kg and from 4.6 to 1.5 mg/kg. The total As content, As(III), and As(V) were all decreased dramatically in brown rice and polished rice. The total As contents in brown rice and polished rice were decreased from 7.9 to 1.6 mg/kg and from 2.3 to 0.68 mg/kg. As(III) concentrations in brown rice and polished rice were decreased from 5.6 to 1.4 mg/kg and from 1.8 to 0.45 mg/kg. As(V) concentrations were decreased from 2.1 to 0.33 mg/kg and from 0.47 to 0.08 mg/kg in brown rice and polished rice. As can be seen, when the concentration of fortified Se reached 1.2 mg/kg, As accumulation was mostly alleviated in the whole grain. This selenium-dependent reduction in As accumulation was also observed in rice hull. The average reductions in rice hull for total As, As(III), and As(V) were 61.5%, 56.4%, and 65.8% when compared with rice treated with As alone. The average reductions in As accumulation in rice bran were 54.4%, 58.4%, and 50.9%; in brown rice 69.1%, 70.3%; and 76.2%, and in polished rice 73.8%, 76.8%, and 78.3%. The reduction in polished rice was obvious. In our study, no DMA and MMA were found in the grains of co-exposure.

Earlier studies have reported the antagonism effect between Se and As, and the addition of Se decreased the total As concentration in rice roots or shoots, which was verified in short-term hydroponic experiments by strengthening the antioxidant potential [34,35]. Ganga Raj Pokhrel et al. investigated the effect of selenium in soil on the toxicity and the uptake and transportation of arsenic speciation to different rice organs from paddy soil [36]. The results revealed that co-addition of As and Se in soils decreased As in grains by 4.76 ± 0.32% and 15.51 ± 0.53% for two cultivars. However, the existence of a good antagonistic effect between Se and the uptake of As in rice plants is still unclear in the case of low-arsenic paddy soil, which highly depends on the rice cultivar, As speciation, and concentration. A similar study also showed this reduction in As accumulation in the root and shoot of rice plants by 14.24% and 23.78%, compared with control [37].

Our data indicated the predominant presence of As(III) in rice bran, brown rice, and polished rice (the average As(III)/As(V) ratio was more than 7:3), although part of As(III) was oxidized to As(V) by the OH radicals in an aqueous environment [38]. Superoxide radicals, hydrogen peroxide, and hydroxyl radicals could accumulate in germinating rice seeds and may promote this oxidation of As(III) to As(V) [39,40].

## 4. Conclusions

In this study, arsenic contents and speciation were analyzed in 38 rice samples collected from two provinces. Selenium biofortification to reduce the risk of arsenic toxicity was researched in germinated rice. Based on the analysis of different parts of rice samples, total As contents in polished rice samples were 0.03–0.37 mg/kg and the mean value of As(III) was nearly twice than that of As(V) in the polished and brown rice, which showed obviously regional variability. The proportion of iAs and the total As in rice bran were higher than those of polished rice, and this was because As tended to accumulate between the husk and the endosperm. Co-exposure to As and Se in germinated rice indicated there was obviously a reduction in As accumulation for total As, As(III), and As(V), especially in polished rice. As(III) was oxidized to As(V) by the OH radicals in the germinated rice. Hence, Se addition may have the potential to alleviate the risk of arsenic toxicity in rice with a positive effect.

## Figures and Tables

**Figure 1 foods-12-02712-f001:**
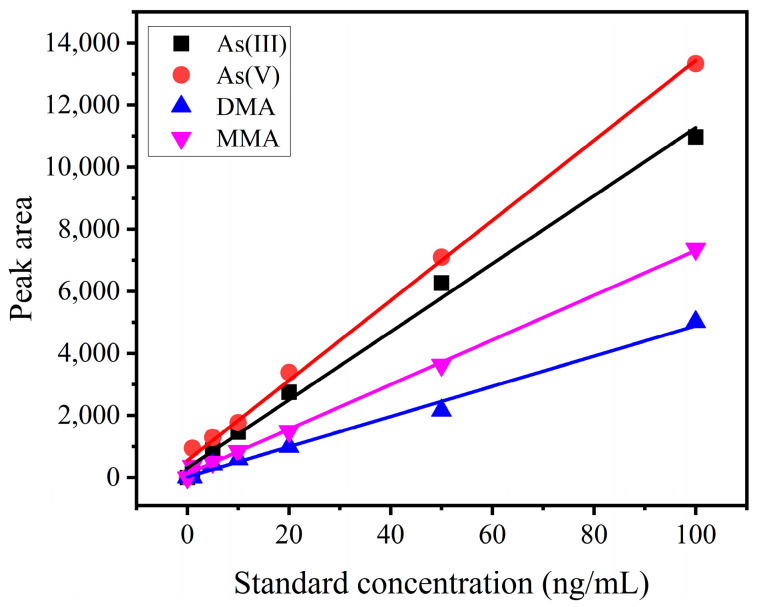
Calibration curves obtained for inorganic arsenic and organic arsenic (As(III), As(V), DMA, and MMA) with HPLC-ICP-MS.

**Figure 2 foods-12-02712-f002:**
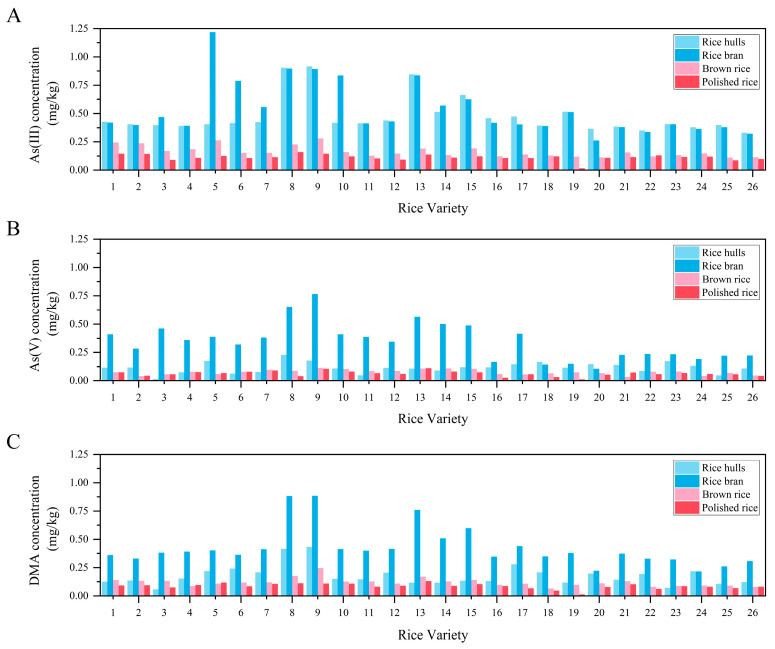
The concentration of As(III) (A), As(V) (B), and DMA (C) in rice hull, rice bran, brown rice, and polished rice from 26 varieties (38 samples).

**Figure 3 foods-12-02712-f003:**
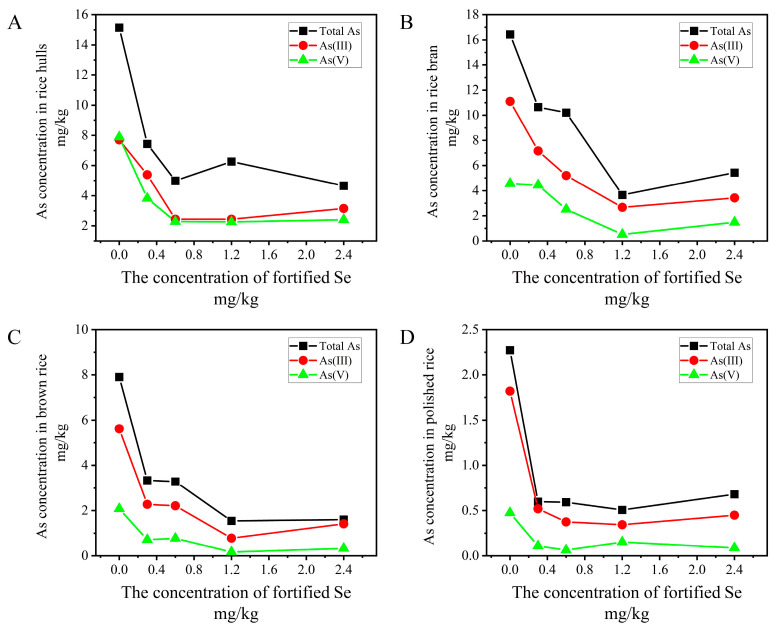
Detection of total arsenic, As(III) and As(V) in different parts of germinated rice ((**A**): rice hull, (**B**): rice bran, (**C**): brown rice, (**D**): polished rice) under selenium stress.

**Table 1 foods-12-02712-t001:** Parameters of the microwave-assisted extraction procedure.

Step	Power (W)	Temperature (°C)	Ramp Time (min)	Hold Time (min)
1	1600	130	10	5
2	1600	160	5	5
3	1600	185	5	5

**Table 2 foods-12-02712-t002:** Parameters of HPLC-ICP-MS for speciation analysis of arsenic.

	Instrument Conditions	Parameters
HPLC	Chromatographic Column	Hamilton PRPX-100 (250 mm × 4 mm)
	Flow Rate	1.0 mL/min
	Injection Volume	50 μL
	Mobile Phase	12.5 mmol/L sodium dihydrogen phosphate buffer, pH = 8.0
ICP-MS	Radio-Frequency Power	1550 W
	Collision Gas Flow Rate	He, 4.0 mL/min
	Carrier Gas Flow Rate	0.85 L/min
	Plasma Gas Flow Rate	15 L/min
	Sampling Depth	8 mm
	Sampling/Skimmer Cone	Nickel, 1.0 mm/0.4 mm
	Scanning Mode	Peak-hopping
	Acquisition Mode	Time-resolved data acquisition
	Dwell Mode	300 ms
	Integration Mode	Peak area
	Isotope Monitored	75 As

**Table 3 foods-12-02712-t003:** Total As contents in rice samples collected from two provinces.

Rice Varieties	Sample Number	Rice Hull (mg/kg)	Rice Bran (mg/kg)	Brown Rice (mg/kg)	Polished Rice (mg/kg)
1–15	19	0.82 (0.46~1.54)	1.58 (1.00~2.53)	0.40 (0.33~0.63)	0.28 (0.21~0.37)
16–26	19	0.68 (0.54~0.89)	0.90 (0.58~1.25)	0.27 (0.23~0.31)	0.21 (0.03~0.28)

**Table 4 foods-12-02712-t004:** Determination results of four arsenic species in rice samples (n = 3).

Rice Varieties	Arsenic Species	Rice Hull (mg/kg)	Rice Bran (mg/kg)	Brown Rice (mg/kg)	Polished Rice (mg/kg)
1–15	As(III)	0.53 (0.39~0.91)	0.65 (0.39~1.22)	0.19 (0.13~0.28)	0.12 (0.086~0.16)
	As(V)	0.11 (0.011~0.022)	0.44 (0.28~0.76)	0.082 (0.037~0.11)	0.070 (0.036~0.11)
	DMA	0.19 (0.055~0.43)	0.50 (0.33~0.88)	0.13 (0.083~0.24)	0.10 (0.071~0.13)
	MMA	-	-	-	-
16–26	As(III)	0.40 (0.33~0.51)	0.38 (0.26~0.51)	0.12 (0.11~0.15)	0.099 (0.012~0.13)
	As(V)	0.12 (0.043~0.17)	0.21 (0.10~0.41)	0.057 (0.03~0.077)	0.045 (0.0085~0.068)
	DMA	0.16 (0.067~0.28)	0.32 (0.21~0.44)	0.091 (0.062~0.126)	0.067 (0.0096~0.099)
	MMA	-	-	-	-

## Data Availability

The data are contained within this article.

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
