# Peer review of "Arsenic Contents, Speciation and Toxicity in Germinated Rice Alleviated by Selenium"

_foods, 2023, doi:10.3390/foods12142712_

Round 1

Reviewer 1 Report

Referee Report

for the manuscript entitledArsenic contents, speciation and selenium 1 alleviates the risk of arsenic toxicity in 2 germinated rice by Zheng et al. that has been submitted to Foods.

This work reports the data in terms of arsenic speciation as well as total arsenic (As) and selenium (Se) levels in a selection of rice samples from the Chinese market.

Although As is a chemical hazard of interest in food chemistry, especially due to the high toxicity of its As(III) and As(V) species, the manuscript in its present state is not of enough scientific quality to be published in the Foods journal. 

A list of general comments is provided below.

  • The title is misleading, it is not clear what alleviates the toxicity in rice
  • The authors made many critical confusions between basic terms in analytical chemistry. Some examples are provided below:

ü  Lines 128-129: “Rice hulls, rice bran, brown rice, and polished rice samples were digested using the microwave-assisted extraction to extract As[17], which were summarized in Table 1.”

ü  Table 1, last column. “Retention time” is a term used in chromatography, not when using a digestion step

ü  Table 2: “Samples size” is nor correct; “injection volume” or ‘sample loop volume” should be used instead

ü  Table 2: “Temperature Of The Atomization Chamber » is totally misleading; I suppose that the authors wanted to say “spray chamber”

ü  Table 2: There is no “Complemental Air” in ICP-MS

ü  Table 2: “Inorganic Arsenic Detection Quality Number”: this term is totally sloppy (invented); the authors must must have simply writen: “detected m/z” or “m/z” (which is the atomic mass of As)

ü  Fig. 1. The authors show calibration curves but they claim that they are chromatograms. It is difficult to understand such a mistake; it looks that the manuscript was not corrected by any senior researcher amongst the (co)authors.

The English and the manuscript’s editing are very poor, which limit its comprehension. Some example are provided below:

ü  Lines 149-150: “The parameters were showed in Table 2.” 

ü  “Based on the detection of 38 rice samples, total As contents in polished rice samples were…”. A sample cannot be detected, it is analysed.

Author Response

The response was listed in attached file. 

Reviewer 2 Report

The submitted manuscript requires major revision. Authors should use the template recommended by the journal. It is necessary to revise the English language by a native speaker or another service offering language corrections. Citations in the text need to be edited; they are often badly formatted. Also, there is always a space missing after the journal title in the list of citations. The abstract is poorly worded, and it should be rewritten. In the experimental part, quality control is not mentioned. The caption for Figure 1 is wrong. It is a calibration curve. The labels in the tables are terminologically incorrect. The authors use the abbreviation iAs (apparently inorganic As) in the text; however, this abbreviation is not described anywhere. On line 251, the authors use the abbreviation for an average. This sentence should be reformulated. There is apparently a typographical error on line 274 (c.>70:30%); it is unclear what the authors mean by this.

English language correction is required.

Author Response

The response was listed in the attached file.

Reviewer 3 Report

The content of the article is interesting from a scientific point of view, it is methodologically well-crafted, but there are some formal flaws in it that would need to be corrected and then also more serious flaws that will be specified further. A formal reminder refers to citing literature in the text - without a space - text[X, Z] - correctly with a space - text [X, Z]; further use of the formulation -"organic and inorganic arsenic (As)" - arsenic is only one and if we want to express its bond, it is necessary to use the formulation "organically bound As" - page 1; on page 4 is expressed E.U. limit for As in rice 200 mg/kg. Today, Commission Regulation (EU) 2023/465 of March 3, 2023, applies, where the limit for not pre-steamed milled rice (polished or white rice) is only 150 mg/kg. Furthermore, in some places in the text, the formulation used (p. 6, 7) is 0.3 ± 0.001. This expression should be corrected to 0.300 ± 0.001.

Among the more serious shortcomings, the chapter "Used instruments" is missing in work, because we have no information about HPLC and some information about ICP-MS is unclear or incorrectly stated: Mode of Acquisition He pattern, Helium flow rate 4.0 mL/min, why He, what role is it? Carrier Gas High purity liquid argon, Ar is used in gaseous form, not liquid; Atomization Chamber - properly Nebulization Chamber; Inorganic Arsenic Detection Quality Number - not clear.

Overall, in the conclusion, it is not clear whether Se was only added in the experiment, or the rice was already fertilized with salts containing Se in the field, and the effect of Se is expressed in a negative context as Se stress, while the effect of Se should be understood positively in relation to As toxicity.

A more exact description of the rice grain and its parts and the effect of Se on the possibility of reducing the concentration in parts of the rice grain.

Author Response

(The authors gave the same response as above.)

Reviewer 4 Report

The authors analyzed different forms of arsenic in rice samples, as well as evaluating the effect of Se on arsenic. 

-The abstract could be improved, especially the objectives part.

-the whole text needs to be revised, as there are duplicate words in various parts of the text

-the authors must provide all data obtained from the validation of the analytical method.

Author Response

(The authors gave the same response as above.)

Round 2

Reviewer 2 Report

The authors responded to the reviewers' comments and made revisions to the manuscript to bring it to an acceptable form. However, the English language revision was not entirely successful, as the text still has linguistically incorrect phrases and sentences.

The level of English in which the text is written is still weak. I recommend that the authors utilize more advanced editing techniques to improve the quality of the professional text.

Reviewer 4 Report

The study is complete and will be of great interest to readers of the journal.